# Thyroid Nodule Characterization: Overview and State of the Art of Diagnosis with Recent Developments, from Imaging to Molecular Diagnosis and Artificial Intelligence

**DOI:** 10.3390/biomedicines12081676

**Published:** 2024-07-26

**Authors:** Emanuele David, Hektor Grazhdani, Giuliana Tattaresu, Alessandra Pittari, Pietro Valerio Foti, Stefano Palmucci, Corrado Spatola, Maria Chiara Lo Greco, Corrado Inì, Francesco Tiralongo, Davide Castiglione, Giampiero Mastroeni, Silvia Gigli, Antonio Basile

**Affiliations:** 1Department of Medical Surgical Sciences and Advanced Technologies “GF Ingrassia”, University Hospital Policlinic “G. Rodolico-San Marco”, 95123 Catania, Italy; giulianatattaresu@gmail.com (G.T.); alessandrapittari@gmail.com (A.P.); pietrofoti@hotmail.com (P.V.F.); spalmucci@unict.it (S.P.); cor_spatola@hotmail.com (C.S.); mariachiaralg@gmail.com (M.C.L.G.); corrado.ini@gmail.com (C.I.); tiralongofrancesco91@hotmail.it (F.T.); davidegiuseppecastiglione@gmail.com (D.C.); basile.antonello73@gmail.com (A.B.); 2Department of Translational and Precision Medicine, “Sapienza” University of Rome, 00185 Rome, Italy; 3Klinika Dani, 1010 Tirane, Albania; ect70@hotmail.it; 4Unit of Radiology, Papardo Hospital, 98158 Messina, Italy; giampieromastroeni@aopapardo.it; 5Department of Diagnostic Imaging, Sandro Pertini Hospital, 00157 Rome, Italy; adrenalina_1@hotmail.it

**Keywords:** ultrasound, thyroid nodules, thyroid cancer, fine-needle aspiration biopsy, thyroid imaging reporting and data system, artificial intelligence

## Abstract

Ultrasound (US) is the primary tool for evaluating patients with thyroid nodules, and the risk of malignancy assessed is based on US features. These features help determine which patients require fine-needle aspiration (FNA) biopsy. Classification systems for US features have been developed to facilitate efficient interpretation, reporting, and communication of thyroid US findings. These systems have been validated by numerous studies and are reviewed in this article. Additionally, this overview provides a comprehensive description of the clinical and laboratory evaluation of patients with thyroid nodules, various imaging modalities, grayscale US features, color Doppler US, contrast-enhanced US (CEUS), US elastography, FNA biopsy assessment, and the recent introduction of molecular testing. The potential of artificial intelligence in thyroid US is also discussed.

## 1. Introduction

Thyroid nodules are a common clinical issue and often pose a diagnostic challenge. Differentiated thyroid cancer, the most frequent subtype of thyroid cancer, is increasing in prevalence [1,2,3]. According to epidemiological evidence, the prevalence of palpable thyroid nodules is higher in women than in men: approximately 5% in women and 1% in men in iodine-sufficient environments [4,5]. The risk of developing a palpable thyroid nodule increases with age, with a lifetime risk estimated at 5–10% [3]. On the other hand, ultrasound (US) has revealed nodules in 19–68% of random thyroid examinations [3]. Despite the high prevalence of nodular disease, malignancy is found in only 7–15% of nodules [6]. The incidence of both thyroid nodules and malignant ones is estimated to be rapidly increasing [6]. This recent increase in prevalence is largely attributed to early discovery through the current use of high-resolution US and detection of small nodules [6,7]. Furthermore, since survival rates for thyroid cancer have remained stable since the advent of US, it is believed that the increased prevalence of thyroid cancer is related to the early detection of small lesions [7]. In South Korea, where there has been a rapid increase in cases of papillary thyroid cancer (PTC) due to the widespread use of thyroid US in asymptomatic patients, mortality rates remain extremely low [8,9]. In the United States, PTC overdiagnosis, defined as the diagnosis of thyroid tumors that would not result in symptoms or death if left alone, accounted for 70–80% of thyroid cancer cases in women and 45% of cases in men between 2003 and 2007 [9,10].

The clinical challenge in the management of thyroid nodules is finding the most appropriate and systematic methodology capable of reliably differentiating benign from malignant nodules in a cost-effective manner. Ultrasound is the primary imaging modality for thyroid nodular pathology. In the past 5–10 years, novel diagnostic and management options have been introduced, ushering in a new era in the diagnosis of thyroid carcinoma. Ultrasound-based thyroid nodule risk stratification systems have been devised by international societies to reduce excessive biopsies, which cause inconvenience to patients and increase public health costs [11]. Additionally, questionable cytology results may sometimes necessitate surgery [11].

Therefore, we decided to provide an overview of the current methods available to minimize unnecessary invasive procedures and treatments.

## 2. Clinical Evaluation and Laboratory Evaluation of Patients with Thyroid Nodules

Early detection of low-risk small PTCs by US has led to overdiagnosis and over-treatment. Therefore, the risk profile of the individual patient and nodule must be considered in diagnostic and management decisions. In managing thyroid nodules, the patient’s history and physical examination significantly influence the probability of malignancy, but they do not provide absolute diagnostic indications.

Characteristics suggesting a malignant diagnosis include the following:Patients younger than 20 or older than 70 years of age;Male patients;Signs of dysphagia or dysphonia;History of neck irradiation;Previous thyroid carcinoma in the same patient;A firm, hard, or immobile nodule upon palpation;Cervical lymphadenopathy upon palpation.

Conversely, factors associated with a benign diagnosis include the following:Family history of autoimmune disease (e.g., Hashimoto’s thyroiditis);Family history of benign thyroid nodule or goiter;Thyroid hormonal dysfunction (hypothyroidism or hyperthyroidism);Nodule that provokes pain or tenderness;A soft, smooth, and mobile nodule upon palpation [12].

Regarding nodule size, US size is considered significant, as nodules larger than 3 cm may have a higher risk of malignancy [12]. However, other evidence indicates that nonpalpable nodules (incidentalomas) found on high-resolution US have a similar risk of malignancy to that of palpable nodules [12].

The multinodular goiter is commonly an additional challenge and the common belief that a multinodular goiter without a dominant nodule indicates benignity is questioned by recent studies involving US-guided fine-needle aspiration biopsy (FNAB) [12,13]. Evidence suggests that the formation of benign nodules and the emergence of malignancies share the same causative factors, as indicated by their increased incidence, particularly with PTCs and benign nodules [13,14]. Thyroid-stimulating hormones (TSHs) are the main mitotic factor, explaining the high incidence of thyroid nodules in endemic goiter regions associated with iodine deficiency [13,14]. Additionally, metabolic syndrome and insulin resistance are linked to a high incidence of nodular goiter and PTC [13].

For laboratory evaluation, a TSH test is warranted to check for hypothyroidism and hyperthyroidism. However, in the majority of thyroid nodule cases, TSH measurements are normal. If TSH is normal in the presence of thyroid nodules, no further laboratory investigation is needed, except when autoimmune disease (e.g., Hashimoto’s thyroiditis or Basedow–Graves) is suspected [12]. In cases of low–normal or high–normal TSH with thyroid nodules, it is mandatory to measure thyroxine (T4) and triiodothyronine (T3) levels [12].

If Hashimoto’s thyroiditis or Basedow–Graves disease is suspected (based on patient and family history, clinical findings, etc.), serum levels of antibodies for antithyroid peroxidase (anti-TPO) and antithyroglobulin (anti-Tg) should be measured [12]. However, a diagnosis of Hashimoto’s thyroiditis can coexist with a diagnosis of a malignant nodule [12]. No other laboratory investigations are valuable in evaluating thyroid nodules [12].

## 3. Imaging

### 3.1. Ultrasound

US is the primary imaging modality in the diagnostic workup of thyroid nodules, and its features are used to determine the need for biopsy with fine-needle aspiration (FNA). Ultrasonographic parameters associated with malignancy with a high predictive value include: microcalcifications (punctate echogenic foci without posterior shadowing), macrocalcifications, a taller-than-wide shape, irregular and spiculated borders, and marked hypoechogenicity [15,16,17]. US findings that favor benign nodules include a large cystic component, a hyperechoic solid nodule, a clear comet tail artifact in punctate foci, and a spongiform appearance [16,17]. Predominantly cystic or spongiform nodules are inherently benign.

Microcalcifications must be distinguished from echogenic foci with comet tail artifacts (not ring down artifacts that originate behind microbubbles trapped in fluid), which is a benign finding due to inspissated crystallized colloids found in microcystic components of a nodule. However, if a punctate echogenic focus in a nodule is not characterized as definitely a colloid, FNA is warranted because microcalcifications are the most specific finding associated with malignancy (~95%) [17,18]. Canon’s MicroPure (Figure 1) has been shown to improve microcalcification detection over grayscale US for assessing echogenic foci (Figure 1) [19]. Microcalcifications are associated with papillary thyroid carcinoma, the most common thyroid malignancy. Both benign and malignant nodules can sometimes present coarse calcifications. PTC and medullary thyroid carcinoma can also have coarse calcifications [18]. Peripheral-rim calcifications can be found in both benign and malignant nodules [18]. A taller-than-wide configuration on US transverse scanning of a nodule is highly suspicious for malignancy [20]. Irregular, lobulated/microlobulated, and spiculated margins are also suspicious for malignancy [18].

Hypoechogenicity in a solid nodule is found in most PTCs and nearly all medullary carcinomas; however, benign nodules can also be hypoechoic. Marked hypoechogenicity, defined as more hypoechoic than anterior neck muscles, is more suspicious. If no other malignant features (e.g., calcifications) are found, hypoechoic nodules are biopsied once they reach size criteria, typically at least 10 mm. Parathyroid adenomas are also markedly hypoechoic and can confound thyroid nodule diagnoses. Parathyroid adenomas are the most common cause of primary hyperparathyroidism, and elevated serum calcium and parathyroid hormone levels aid in differentiating them from suspicious thyroid nodules [21]. Isoechogenicity is associated with benignity, but follicular carcinomas and some medullary carcinomas can be isoechoic [17,18]. Large cystic components in a nodule favor a benign diagnosis, although a significant proportion of PTCs can have cystic components [17,18].

The halo sign around a well-marginated hypoechoic or isoechoic nodule is frequently encountered in benign nodules, principally follicular adenoma [17,22]. However, more than 50% of benign nodules do not present this sign [21]. Conversely, 24% of papillary carcinomas have a halo, whether complete or incomplete [21]. A characteristic feature of anaplastic thyroid carcinoma and thyroid lymphoma is the extrathyroid invasion of adjacent structures [17].

Cervical lymph nodes should be routinely scanned in all thyroid US studies, and enlarged lymph nodes should be considered suspicious for thyroid malignancy, especially PTC [3]. Typical signs of lymph node metastatic invasion include microcalcifications, macrocalcifications, cortical hyperechogenicity (focal or diffuse), and lymph nodes with cystic changes [23]. Suspicious signs include the loss of the normal fatty hilum and an irregular node appearance. Highly suspicious signs include irregular increased color Doppler flow with loss of the normal hilar in–out flow and the appearance of peripheral in–out blood flow. While there are no definite threshold criteria for lymph node biopsy in the literature, generally, a biopsy is warranted if suspicious features are combined with a lymph node size greater than 8 mm [17,18,20,22]. For thyroid nodules, no US features are 100% sensitive or specific, but regional lymphadenopathy with microcalcifications is 100% specific [18,20].

### 3.2. Color Doppler US

Regarding color Doppler US (CDUS) and power Doppler for thyroid nodules, despite initial optimism and promising data, recent studies have shown a limited role of CDUS in diagnosing thyroid cancer. Frates et al. found that solid hypervascular thyroid nodules have a high likelihood of malignancy (42%), but they noted that the color characteristics of a thyroid nodule cannot be used to exclude malignancy because 14% of solid non-hypervascular nodules were malignant [24]. According to Moon, vascularity alone, or a combination of vascularity and grayscale US features, was not as useful as using suspicious grayscale US features alone in predicting thyroid malignancy [25]. In a large 2020 study, CDUS was used in combination with and compared to the American College of Radiology (ACR) TI-RADS classification. The researchers found that CDUS did not improve the risk stratification ability of the ACR TI-RADS US system [26].

An innovation in US, Superb Microvascular Imaging (SMI) by Canon, is capable of canceling background artifacts and noise in the image, thus cleaning the Doppler motion detection and enhancing the vascular signals. SMI displays images comparable to CEUS without using any intravenous contrast agents. SMI can depict the slowest blood motion in small vessels and is therefore superior to color Doppler US [27]. This emerging US technique may offer an advantage in assessing malignant thyroid nodules, although the data in the literature are still limited.

### 3.3. Contrast-Enhanced US

Contrast-enhanced US (CEUS) utilizes a microbubble contrast agent to depict real-time microvascular flow, allowing for dynamic visualization and quantification of all thyroid vascularization down to the smallest capillary level. The microbubble contrast agent does not interfere with thyroid function. Currently, there are no universally accepted standards for quantitative or qualitative CEUS studies of thyroid nodules, and no single feature of CEUS appears to be sensitive and specific enough for diagnosing malignancy. CEUS is used in clinical research, and several studies have evaluated its diagnostic accuracy in distinguishing malignant thyroid nodules from benign ones. However, due to high variability in sensitivity and specificity, the results remain controversial, and its value is not yet clear. Furthermore, the cost effectiveness of CEUS remains a major impediment to its widespread use.

A literature review [28] found no clear utility of CEUS in the differential diagnosis of nodules. However, CEUS can play a potential operative role in thyroid ablation procedures for nodules and lymph node metastases. Additionally, CEUS can be used to evaluate the radicality of thyroid surgery, survey for malignancy relapse at the margins of ablated thyroid areas, and follow up on progressive changes in necrotic regions during the post-treatment period [28].

### 3.4. Elastography

US elastography is an emerging variation of ultrasound imaging. Although the technique is still being developed for clinical diagnosis, its utility remains questionable in various applications. There are two forms of elastography: strain elastography and shear wave elastography.

Strain elastography (Figure 2) is based on the operator’s free-hand compression of the tissue to detect strain in the axial dimension, characterizing the tissue’s elasticity. This is performed by comparing the physical features of the US beam from the tissue before and after compression. Some manufacturers offer a semi-quantitative evaluation of tissue strain in their US devices by comparing the elasticity of a nodule with that of the adjacent normal thyroid parenchyma [29,30].

Shear wave elastography is based on a concept similar to strain elastography. Instead of using transducer pressure to measure changes in strain, an intense ultrasound pulse is transmitted to produce lateral shear waves. These shear waves are then tracked with low-intensity pulses to determine shear velocity, from which elasticity is calculated in kPa [30,31].

The rationale for using US elastography in thyroid nodule examinations is that malignant nodules have increased stiffness due to the high content of collagen and myofibroblasts [30,32]. However, several studies have provided contradictory results on the sensitivity and specificity of both types of elastography compared to conventional US and US classification systems [30,33,34,35]. Additionally, factors such as the structural heterogeneity of thyroid nodules and operator-dependent variables, including pre-compression and selection of the scanning plane, impede the accuracy of the elastographic technique [30].

An extensive literature review found that US elastography does not sufficiently fulfill its initial promise of increasing the accuracy of US in differentiating thyroid nodules. Consequently, it has not become a clinically reliable tool, and grayscale US nodule features remain the mainstay in clinical practice [30].

In conclusion of this presentation so far, the authors would like to stress that the classical grayscale US imaging features of suspicious versus benign and non-suspicious nodules, even nowadays, remain the mainstay of thyroid nodule diagnosis and characterization, while the new methods find niche applications. Advanced equipment and in-depth knowledge for avoiding pitfalls in diagnoses are required by dedicated specialized US radiologists and practitioners in order to utilize US methods other than the classical grayscale US nodule features. Moreover, contrast-enhanced US implies the conspicuous additional costs of the contrast microbubble agent.

## 4. US Classification Systems

Risk classification models based on formal assessment criteria of US features have been created by multiple professional societies to determine the need for FNA. Their purpose is to address concerns regarding variation in practice and overdiagnosis of clinically insignificant thyroid malignancies [17]. These classifications have devised a lexicon standard for nodule features to produce a numeric scoring of US nodule features. The end result of these models is the generation of a numeric score, which categorizes the relative risk for benignity or malignancy and specifies recommendations for further nodule management. The foremost objective is the reduction in unnecessary FNA and unwarranted US surveillance [36].

Studies have shown that the use of approved terminology, following the recommendations of a classification system, results in pertinent, standardized, and reproducible reports, offering clarity to colleagues and clinicians and avoiding misinterpretation [36]. The use of approved classification criteria has been shown to result in a meaningful reduction in the number of thyroid nodules recommended for biopsy and to significantly improve the accuracy of recommendations for nodule management [37].

Among the main classification systems, those of the American College of Radiology (ACR), Korean Society of Thyroid Radiology, and European Thyroid Association seem to correlate better with cytological and histological findings [38,39]. Other main systems include the American Thyroid Association (ATA) classification and the Society of Radiologists in Ultrasound (SRU) guidelines, which are also regularly used.

### 4.1. The Thyroid Imaging Reporting and Data System of the American College of Radiology (ACR TI-RADS)

The ACR TI-RADS score is defined by the following US features: composition, echogenicity, shape, margins, and echogenic foci; the higher the score, the higher the probability of malignancy [40]. The US features are scored as follows [9]:Composition: cystic or completely cystic, 0 points; spongiform, 0 points; mixed cystic and solid, 1 point; solid or almost completely solid, 2 points.Echogenicity: anechoic, 0 points; hyper- or isoechoic, 1 point; hypoechoic, 2 points; very hypoechoic, 3 points.Shape (assessed on the transverse plane): wider than tall, 0 points; taller than wide, 3 points.Margins: smooth, 0 points; ill-defined, 0 points; lobulated or irregular, 2 points; extrathyroidal extension, 3 points.Echogenic foci: none, 0 points; large comet tail artifacts, 0 points; macrocalcification, 1 point; peripheral or rim calcifications, 2 points; punctate echogenic foci, 3 points [9].

When assessing a nodule, the reader selects one feature from each of the first four categories and all applicable features from the fifth category and sums the points (TR). The total points determine the nodule’s ACR TI-RADS level, ranging from TR1 (benign) to TR5 (high suspicion of malignancy) [9]. A sixth feature, size, is used to determine the need for FNA or follow-up US. In cases of multiple nodules, the ACR system recommends scoring the four nodules with the highest scores (not necessarily the largest), and these should be followed up [9]. TR1 is allocated to predominantly cystic and predominantly spongiform nodules as these are inherently benign, and no further points are assigned to them [9].

The ACR TI-RADS recommendations are as follows:TR1: no FNA;TR2: no FNA;TR3: if ≥1.5 cm, follow-up; if ≥2.5 cm, FNA; follow-up US at 1, 3, and 5 years;TR4: if ≥1.0 cm, follow-up; if ≥1.5 cm, FNA; follow-up US at 1, 2, 3, and 5 years;TR5: if ≥0.5 cm, follow-up; if ≥1.0 cm, FNA; US follow-up every year for the next 5 years.

An FNAB is required for suspicious nodules (TR3-TR5) that meet the size criteria mentioned above. Interval enlargement on follow-up is defined as meaningful if there is an increase of >20% in two dimensions and at least >2 mm in each dimension or if there is a >50% growth in volume [40]. If the ACR TI-RADS score increases in a follow-up US, an interval scan the following year is again required [40]. The scientific evidence for malignancy for each TR score is as follows: TR1: 0.3%; TR2: 1.5%; TR3: 4.8%; TR4: 9.1%; and TR5: 35% [41].

In a prospective analysis of 502 nodules, Grani et al. found that ACR TI-RADS prevented unnecessary biopsies in more nodules compared to other systems [42]. Two research studies have demonstrated for the ACR system a sensitivity of 75–97% and a specificity of 53–67%, which constitutes the highest sensitivity but also the lowest specificity compared to other scoring classifications [40,43]. However, other studies have found, conversely, the highest specificity for the ACR system [40,42,44]. In these two studies, the ACR system showed the highest performance as measured by the area under the receiver operating characteristic (ROC) curve and the highest accuracy, resulting in the lowest rates of unnecessary FNABs (lowest false positives) [40,42,44]. Another study found a good Cohen kappa of 0.61, demonstrating high interobserver agreement regarding the decision to send a patient for an FNAB [45].

### 4.2. American Thyroid Association (ATA) Management Guidelines for Adult Patients with Thyroid Nodules and Differentiated Thyroid Cancer

As they have performed periodically in the past, during 2015, an ATA task force worked on and prepared detailed guidelines with evidence-based recommendations, which were published in 2016 in an authoritative 133-page paper. These guidelines cover the initial evaluation of nodules, clinical and US standards for FNABs, FNAB result interpretations, the introduction of molecular markers, and the management of benign nodules. Recommendations for thyroid cancer diagnosis are also included, detailing the screening procedure, the staging of thyroid cancer, and risk evaluation [3].

According to the ATA guidelines, a nodule is classified into one of five US feature categories, with the following indications for an FNA biopsy:Benign US pattern (0% risk of malignancy): no FNAB required;Extremely low-suspicion US pattern (<3% risk of malignancy): FNAB if ≥2 cm (or US follow-up);Low-suspicion pattern (5–10% risk of malignancy): FNAB if ≥1.5 cm;Intermediate-suspicion pattern (10–20% risk of malignancy): FNAB if ≥1 cm;High-suspicion pattern (>70–90% risk of malignancy): FNAB if ≥1 cm.

In the ATA guidelines, the patterns are defined on US with the following features:Benign pattern (0% risk of malignancy): completely cystic nodules with fine walls;Extremely low-suspicion pattern (<3% risk of malignancy): spongiform nodules and nodules with interspersed cystic spaces and no features of the higher-level patterns;Low-suspicion pattern (5–10% risk of malignancy): isoechoic or hyperechoic nodules, or partially cystic nodules with a peripheral solid component, and no features of the higher-level patterns;Intermediate-suspicion pattern (10–20% risk of malignancy): hypoechoic solid nodules with smooth margins and no features of the higher-level patterns;High-suspicion pattern (>70–90% risk): solid hypoechoic nodules or solid hypoechoic components of partially cystic nodules, with at least one of these features: microcalcifications, irregular margins (infiltrative, microlobulated), extrathyroidal extension, a taller-than-wide shape, rim calcifications with an extrusive soft tissue component, or lymphadenopathy.

The ATA guidelines consider coarse intranodular macrocalcification and rim calcifications as US features implying an increased risk for malignancy but to a lesser degree than microcalcifications.

A study using the ATA guidelines in 604 nodules [46] questioned the ATA size cutoffs for FNA biopsy, which are as follows: low risk at 15 mm and intermediate risk and high risk at 10 mm. Based on their findings, the authors suggest that it would be clinically safe to increase the ATA cutoffs for FNA in low-risk nodules to 25 mm and in intermediate-risk nodules to 15 mm, which will conserve the high performance of NPV (negative predictive value) while improving accuracy and sparing unnecessary FNA.

### 4.3. Korean Society of Thyroid Radiology (KSThR): Thyroid Imaging, Reporting and Data System (K-TIRADS)

The KSThR published its K-TIRADS criteria in 2016 and revised them in 2021 and 2023, defining recommendations for the US lexicon, biopsy criteria, US criteria of extrathyroidal extension, optimal thyroid computed tomography (CT) protocol, and US follow-up of thyroid nodules before and after biopsy [23,47,48]. KSThR categorizes nodules into five K-TIRADS according to the following US criteria:K-TIRADS 1: No nodule.K-TIRADS 2: Benign category:
Iso-/hyperechoic spongiform;Partially cystic nodule with intracystic echogenic foci and comet tail artifact;Pure cyst malignancy risk is <3%. No biopsy indicated.K-TIRADS 3: Low-suspicion category: Partially cystic or iso-/hyperechoic nodule without any of the three suspicious US features. Malignancy risk is 3–10%. Biopsy indicated at >2 cm.K-TIRADS 4: Intermediate suspicion:
Solid hypoechoic nodules with no other suspicious US pattern;Partially cystic or iso-/hyperechoic nodule with any of the three suspicious US characteristics;Completely calcified nodule malignancy risk is 10–40%. Biopsy indicated at >1 or 1.5 cm.K-TIRADS 5: High suspicion: Solid hypoechoic nodule with any of the three suspicious US features (punctate echogenic foci, nonparallel orientation, and irregular margins). Malignancy risk is >60%. Biopsy indicated at >1 cm.

### 4.4. European Thyroid Association TIRADS (EU-TIRADS)

EU-TIRADS [49] classifies nodules into five categories: EU-TIRADS 1 indicates an absence of nodules; EU-TIRADS 2 denotes benign nodules (cystic and microcystic); EU-TIRADS 3 signifies low-risk nodules (oval shape, smooth margin, iso/hyperechoic, and without high-risk patterns); EU-TIRADS 4 categorizes intermediate-risk nodules (oval shape, smooth margin, mildly hypoechoic, and without high-risk characteristics); and EU-TIRADS 5 identifies nodules with any of the high-risk US characteristics (non-oval shape, irregular margin, microcalcifications, and marked hypoechogenicity). Nodules are sent for an FNAB if they meet these criteria: EU-TIRADS 2: no FNAB (unless for therapeutic purposes, to relieve compression, or for cosmetic reasons); EU-TIRADS 3: FNAB if >20 mm; EU-TIRADS 4: FNAB if >15 mm; EU-TIRADS 5: FNAB if >10 mm; consider an FNAB or US follow-up if <10 mm. EU-TIRADS recommends that a lesion hot on scintigraphy be considered benign, and no FNAB is required regardless of its US features.

## 5. Secondary Imaging Methods

### 5.1. Thyroid Scintigraphy: Nuclear Imaging Study of the Thyroid

Nuclear scintigraphy is commonly used for evaluating physiologic thyroid function and identifying metabolically active and inactive nodules. However, routine workup of thyroid nodules nowadays does not typically include scintigraphy [12]. Nevertheless, scintigraphy can be useful in categorizing nodules based on radioactive isotope uptake as hot, warm, or cold. Hot nodules typically function autonomously, warm nodules are associated with normal thyroid function within the nodule, and cold nodules indicate hypo- or nonfunctioning thyroid nodular components [12]. Hyperfunctioning hot thyroid nodules generally have a high negative predictive value for malignancy. Conversely, 5–8% of warm and cold nodules are malignant [12,50].

Scintigraphy with ultrasound was routinely performed in the past as the primary method for evaluating thyroid nodules. However, this approach has since been shown to result in a low yield of cancer diagnoses, largely because less than 10% of solitary thyroid nodules are hot and approximately 90% of cold nodules are benign [12].

### 5.2. Thyroid Computed Tomography (CT)

CT is primarily required for evaluating thyroid tumors that are substernal. Thyroid pre- and post-contrast scans are performed using an optimized dedicated protocol. Pre-contrast scans are used to visualize calcifications and ectopic thyroid tissue and aid in the differential diagnosis between tumor recurrence and residual thyroid tissue after thyroidectomy [23]. Post-contrast scans, among other purposes, serve to detect lymph node metastases. They reveal lymph nodes as nodes with heterogeneous enhancement, with some parts showing strong-contrast uptake and other parts showing weak or no uptake, as well as cysts [23].

## 6. FNAB

A fine-needle aspiration biopsy (FNAB) is crucial in diagnosing thyroid malignancies. Specimen adequacy and diagnostic accuracy vary due to several factors; however, a body of evidence has shown the high accuracy of FNABs, with their sensitivity above 80% and specificity above 90% [12,51]. Nonetheless, FNAB accuracy varies among different centers due to inconsistencies and variability in the expertise and experience of cytopathologists in thyroid conditions. Moreover, it is further complicated by the quality of the biopsy sample dependent on the technical skills and experience of the operator performing the FNAB [12].

In a 2007 conference, the National Cancer Institute established a consensus for diagnostic classification to categorize specimens clearly for effective communication and cytopathology description. This led to improved result communication and clarity in thyroid cytopathology diagnosis. Known as the Bethesda System for Reporting Thyroid Cytopathology, this consensus terminology assigns FNAB results into categories indicating different risks of malignancy as follows: benign, <1%; atypia of undetermined significance, (AUS) 5–10%; follicular neoplasm, 20–30%; suspicious for malignancy, 50–75%; and malignant, 100% [12,52].

Category 3 of the Bethesda System includes “atypia of undetermined significance/follicular lesion of undetermined significance” [53], which can complicate patient management. An Italian consensus for thyroid cytopathology classification has much similarity with the Bethesda System; however, it introduces a subclassification of indeterminate for malignancy, the TIR-3 category, further divided into TIR-3A (low-risk) and TIR-3B (high-risk). This distinction aims to differentiate between probably benign nodules (high negative predictive value) and malignancies (high sensitivity), though it offers limited value for adenomas (modest specificity, low positive predictive value). The Italian classification enhances sensitivity compared to the Bethesda System [54].

Ultrasound (US) is typically used to guide FNABs, ensuring pathologists obtain high-quality specimens to assign accurate diagnoses and minimize the need for repeat FNABs. Adequate sampling during FNABs relies on meticulous technical preparation and the experience of the operator performing the US-guided procedure [12].

A common challenge with FNAB samples includes obtaining hypocellular specimens or materials with high follicular cellularity [12,52]. Biopsy techniques often target cystic and microcystic nodules, resulting in hypocellular aspirates. To mitigate these issues, some institutions involve cytotechnologists or pathologists to assist during FNABs, verifying sample adequacy on-site immediately after collection [12,52].

Pathologists may struggle to differentiate benign follicular neoplasms from malignancies when faced with samples of high follicular cellularity. Moreover, the evaluation is complicated by Hurthle cells which are larger than typical follicular cells and have abundant mitochondria. Specimens showing dense Hurthle cellularity may indicate benign or malignant Hurthle cell tumors or Hashimoto thyroiditis [12,55]. Advances in cytologic assessment, including immunocytochemical studies and genetic/molecular profiling of aspirates, aim to enhance FNAB accuracy [12,56].

To address inconclusive results and indeterminate interpretations from FNABs, core biopsy is recommended. Core needle biopsy follows non-diagnostic or indeterminate FNABs, providing larger tissue samples for architectural and cellular detail, facilitating ancillary testing. Increasingly, researchers advocate core biopsy over FNABs for its effectiveness and safety, though local pain and bleeding risks may occur in some cases [57].

### Post-FNAB Management of Thyroid Nodules

According to the Bethesda System, surgery is warranted for the following categories: follicular neoplasm, suspicious for malignancy, and malignant classifications. Since 20–30% of follicular cytopathologies are malignancies, these cases are referred for surgical consultation. However, malignant lymphoma diagnoses do not typically involve surgery, and anaplastic carcinoma usually does not benefit from surgical intervention [12,58].

Management decisions for cases of atypia of undetermined significance include (a) a follow-up FNAB at 3–6 months; if the second US-guided FNAB result remains atypical, surgery is recommended. (b) Surgery may be recommended if, in addition to atypia, suspicious US features are present: a taller-than-wide shape, marked hypoechogenicity, irregular borders, and microcalcifications [12,58].

Nodules classified as benign can be safely monitored with US every 6 to 18 months to assess for changes in characteristics or significant growth. Some institutions require a second confirming FNAB 6–12 months after a benign diagnosis, despite evidence showing low false-negative rates with FNABs. Additionally, it is common practice to repeat an FNAB if US features of the lesion change or become suspicious during follow-up [12,58]. In cases of non-diagnostic FNAB results, the decision may involve repeating an FNAB or opting for core needle biopsy.

## 7. Advancements in Cytologic Analysis

### Molecular Testing

To address the diagnostic challenges posed by thyroid nodules with indeterminate results from fine-needle aspiration (FNA) biopsy, modern approaches propose further supportive testing based on advances in cytologic analysis. These tests, known as “ancillary molecular testing”, utilize commercial panels that assess multiple molecular alterations. Current commercialized molecular tests for thyroid FNA examine mutations, gene fusions, gene expression alterations, microRNA expression, chromosomal copy number alterations, or combinations thereof [59].

Ancillary molecular testing is increasingly integrated into the clinical management of patients with indeterminate FNA cytology, as evidence suggests they enhance diagnostic accuracy and optimize patient care [3,60]. These molecular tests are particularly applicable in Bethesda category III (atypia of undetermined significance or follicular lesion of undetermined significance) and Bethesda category IV (follicular neoplasm or suspicious for follicular neoplasm). Additionally, molecular testing in Bethesda category V (suspicious for malignancy) can aid in deciding between total thyroidectomy and thyroid lobectomy in specific cases [3,61].

Most recent research and clinical applications focus on three commercially available molecular tests: ThyroSeq v3, ThyGeNEXT/ThyraMIR (MPTX), and Afirma Gene Sequencing Classifier. The Afirma Gene Sequencing Classifier (Veracyte, Inc., South San Francisco, CA, USA) uses algorithmic mRNA expression analysis; ThyroSeq v3 Genomic Classifier comprehensively assesses various cancer-related genetic alterations; and ThyGeNEXT/ThyraMIR (Interpace Biosciences, Inc., Parsippany, NJ, USA) evaluates a limited panel of gene mutations and fusions alongside the expression of 12 microRNAs [61].

The Afirma Gene Expression Classifier assists in decision making regarding surgical recommendations for aspirates with cytology of indeterminate significance or follicular lesions [12,56]. Specifically, the Afirma microarray-based Gene Expression Classifier (GEC) exhibits high sensitivity and negative predictive value (NPV), effectively ruling out malignancy in samples with indeterminate cytology. However, its low specificity and positive predictive value (PPV) limit its ability to confirm malignancy [62]. In contrast, the Afirma Gene Sequencing Classifier (GSC), as demonstrated in meta-analyses, reduces unnecessary surgical interventions and facilitates tailored clinical decisions for patients with indeterminate FNAB results [62].

Another ancillary multiplatform genetic test for indeterminate FNAB cytology is MPTX (ThyGeNEXT/ThyraMIR), which combines a mutation panel and a microRNA risk classifier to provide three diagnostic categories (negative, moderate, and positive). Research, including blinded multicenter studies, indicates that MPTX can influence nodule management and surgical decisions significantly. MPTX demonstrated 95% sensitivity and 90% specificity for nodules classified as Bethesda III and IV cytology [63].

## 8. Artificial Intelligence (AI) Systems

Radiomics is an advanced image analysis technique involving automated extraction of extensive quantitative metrics, known as radiomic features, from standard-of-care medical images. These features capture quantitative characteristics of tissue and lesions, such as heterogeneity and shape, performed rapidly and reproducibly by software. The data from radiomic features are analyzed using AI systems: machine learning (ML) or deep learning (DL). This information can provide diagnostic, prognostic, and/or predictive models, contributing to precision medicine [64].

The results of a radiomics analysis can be integrated into Computer-Aided Diagnosis (CAD) software, which offers clear visualization for easy and direct use by clinicians in differential diagnoses of nodules. CAD software may function as a standalone application on a workstation or be fully integrated into ultrasound devices [65]. Research has demonstrated that AI improves sonographic accuracy for thyroid nodule diagnoses and reduces interobserver variability [65]. Initial applications of radiomics in thyroid nodules focused on analyzing texture and ultrasound intensity, which can be influenced by equipment parameters such as gain, dynamics, operator dependency, probe variability, and equipment performance [65].

A recent review of 166 studies on AI for thyroid nodule diagnoses found high accuracy rates (>90%) across all the studies analyzed [65]. Overall, AI accuracy was comparable to that of expert radiologists; however, nearly half of the studies reported higher diagnostic accuracy with AI compared to expert radiologists, while the other half favored expert radiologists. The review concludes that current AI evaluations for thyroid nodule diagnoses are comparable or slightly inferior to those of expert ultrasound specialists and radiologists. Therefore, AI diagnostic outputs require supervision by radiologists, and these promising results suggest AI solutions serve as supportive tools. Additionally, current AI systems are applicable for educational purposes in academic settings.

Research in radiomics is rapidly expanding with numerous publications and commercial applications; however, it is also accompanied by scientific challenges and overly enthusiastic claims by researchers and clinicians alike [64]. While the future prospects for radiomics appear promising, efforts are needed to standardize and validate these techniques to bridge the gap between fundamental research and clinical practice.

In the future, AI systems, together with the advancements in US technology, will change the role and work routine of radiologists and US specialists [64,65], enhancing efficiency and accuracy. However, complex evaluation from the US specialist will always be necessary for integrating imaging findings with clinical data, patients’ history, and diagnostic tests. The advanced diagnostic skills, critical thinking, and specialized knowledge of US specialists will always be essential in handling complex and atypical US nodular presentations and patients that AI alone will not be fully equipped to deal with. Ethical and legal responsibility, interdisciplinary collaboration, continual learning and adaptation, and also patient interaction and communication will always remain human skills.

## 9. Conclusions

Collaboration among experienced and dedicated clinicians, radiologists, and pathologists, guided by comprehensive knowledge of classification systems, is crucial for diagnosing and making decisions regarding thyroid nodules.

Understanding and correctly applying ultrasound (US) risk classifications, particularly TIRADS, are fundamental for initiating optimal management of patients with thyroid nodular pathology.

## Figures and Tables

**Figure 1 biomedicines-12-01676-f001:**
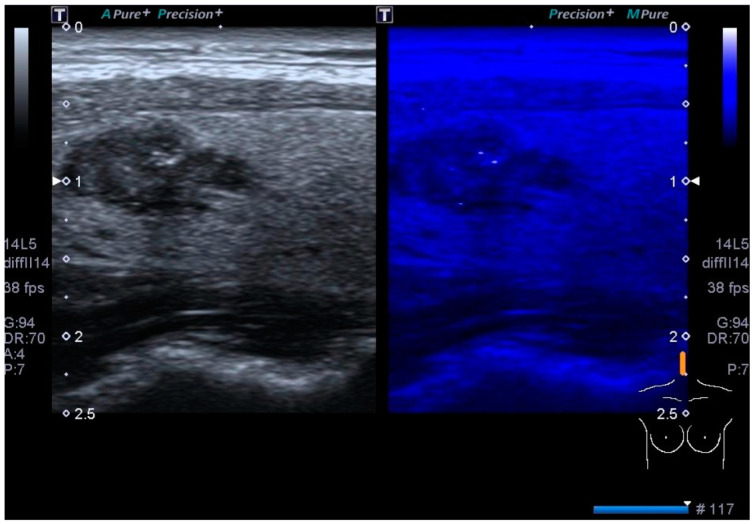
The image shows microcalcifications detected with Micropure (Canon) in the context of histologically proven papillary carcinoma. On the right image, the punctate foci that are truly macrocalcifications and not inspissated crystallized colloids, shine up, and are conspicuous.

**Figure 2 biomedicines-12-01676-f002:**
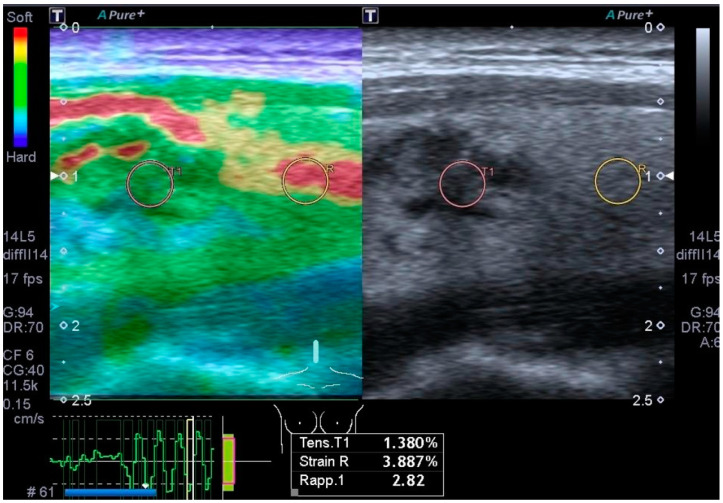
The image shows an evaluation of the strain ratio of a histologically proven malignant tumor, which is slightly higher than the threshold values considered in the literature. The strain ratio represents a semi-quantitative assessment of the stiffness of the nodule under examination, in comparison with the adjacent healthy thyroid tissue by affixing two regions of interest (ROI). Currently, the literature provides us with Canon equipment with a cutoff of 2; in this case, however, the strain ratio showed a value of 2.82.

## Data Availability

The data presented in this study are openly available at https://pubmed.ncbi.nlm.nih.gov/.

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
