# Peer review of "Thyroid Nodule Characterization: Overview and State of the Art of Diagnosis with Recent Developments, from Imaging to Molecular Diagnosis and Artificial Intelligence"

_biomedicines, 2024, doi:10.3390/biomedicines12081676_

Round 1

Reviewer 1 Report

Comments and Suggestions for Authors

Emanuele et al. proved the diagnostic guideline on thyroid nodules (TNs). Too many text for billion times described US, scinti, FNAB, and TIRADS. But, the analysis of novel methods, molecular testing and AI, is so thin and need paper reorganization: the shrinkage of previous diagnostic presentation and the extension of molecular biology methods and AI.

Comments on the Quality of English Language

English need to be refined by native speaker. Please, fix the grammar errors.

Author Response

Answers to reviewer 1:

As the reviewer asked, we corrected the language with a native English speaker for refinement, grammar errors and superfluous commas as the different reviewers asked us.

We added a large paragraph at the end of AI novelties as the reviewer asked. However it is our humble opinion that we did not attempt to bring only the new developments but to present a comprehensive overview of thyroid nodules characterization, that will serve to the reader to refresh and update his/her knowledge.

Reviewer 2 Report

Comments and Suggestions for Authors

The author summarized a comprehensive overview of the current state of diagnosis and recent developments in the characterization of thyroid nodules. It covers various imaging modalities, including ultrasound (US), and discusses the diagnostic value of these modalities in reducing unnecessary invasive procedures and treatments. The review also discusses US classification systems for thyroid nodules. ​ The review also provides information on molecular testing and the use of artificial intelligence (AI) systems in the diagnosis and management of thyroid nodules. ​

Comments and suggestions:

1.    Provide more details about Figure 1 and Figure 2. There is no description in the main text or figure caption.  

2.    What’s the advantage of each imaging method? It’s better to summarize the advantage in the end of section 3.

The review is well-referenced, with citations to relevant studies and literature. In summary, I recommend accepting after minor revision.

Author Response

Answers to reviewer 2:

Figure 1 and Figure 2 were provided with more details and integrated at the right place in text as the reviewer rightly asked. 

At the end of Section 3 as the reviewer 2 correctly asked, we added a summarization and conclusion paragraph for what presented that far in our paper.

Reviewer 3 Report

Comments and Suggestions for Authors

In this overview article, the authors presented a comprehensive description of the clinical evaluation and laboratory evaluation of patients with thyroid nodules employing different imaging modalities, among them: gray scale US features, color US Doppler, Contrast-Enhanced US, US elastography, fine-needle aspiration (FNA) biopsy assessment, also they presented introduction of the molecular testing. The authors concluded that for nodules diagnosis, AI evaluation results are similar or inferior to expert US specialists.

Comments:

1)     This reviewer thinks that, for this overview with description state-of-the-art in diagnostics that use different imaging technique employing AI, the authors should present an additional section where it could be resumed and characterized different approaches, presenting also practical recommendations.

2)     Please correct grammar errors: the word “features”, line 20, and in phrase “however a body of evidence has shown the high accuracy…”, lines 393-394. Grammar and stylistic errors (mainly commas) can be observed in the text of this manuscript. This reviewer suggests revising the text by native speaker that can correct the gramma and stylistic errors.

Comments on the Quality of English Language

Grammar and stylistic errors (mainly commas) can be observed in the text of this manuscript. This reviewer suggests revising the text by native speaker that can correct the gramma and stylistic errors.

Author Response

Answers to reviewer 3:

The reviewer 2 asked for a paragraph with practical recommendations that characterize different approaches. For that, we added a summarization and conclusion paragraph with practical advice, at the end of Section 3, as the reviewer rightly required.

As the reviewer asked, we corrected the language of the whole text, with a native English speaker for refinement, grammar errors, style and superfluous commas. All the errors that the reviewer pointed out were corrected.

Reviewer 4 Report

Comments and Suggestions for Authors

1. Introduction
33 - Could you please split the first sentence into two for a better understanding of the content of the text.
General comment - Could you please structure the introduction from more general to more specific. Start with epidemiology, types of nodes, diagnostic methods and cancer as the most serious pathology and diagnostic challenge.

2. Clinical Evaluation and Laboratory Evaluation of Patients with Thyroid Nodules
66 - Could you please split the first sentence into two for a better understanding of the content of the text.
81 - The sentence about multinodular thyroid goiter is unrelated to the previous paragraph. Present it as a diagnostic challenge and continue on about it.
91 - Perhaps for better understanding you should put this section before the section on multinodular thyroid goiter.

3. Imaging
120 - Can you please explain "ring down artifact" or please leave it out
130 - Maybe it would be better to first write and describe the malignant and benign morphological characteristics of thyroid changes/nodeles and then describe the types of calcifications

4. US Classification Systems
238 - Did you mean "is generating"
252 - regularly used
303 - Unclear meaning of the first part of the sentences.
General comment - Korean Society of Thyroid Radiology and European Thyroid Association TIRADS. You did not mention some studies and experiences with the above classifications.

5. Secondary Imaging Methods - all good

6. FNAB
395 - the sentence is confusing, please split it into two
407 - You mention the Italian consensus without first explaining it. First explain that system and then compare it with the Bethesda system.
427 - you mention Hurthe cells, but you did not explain which cells they are
443 - the sentence about lymphoma and anaplastic carcinoma is unrelated to the previous sentences. Incorporate them differently into the text.

7. Advancements in Cytologic Analysis - all good

8. Artificial Intelligence (AI) Systems - all good

9. Conclusion – all good

Comments on the Quality of English Language

moderate editing

Author Response

Answers to reviewer 4:

Introduction: The sentence was split down as asked. A restructure of the introduction to make it more clear and fluent for the reader, as the reviewer asks, was performed, starting with a broad overview of the prevalence and challenges associated with thyroid nodules, gradually narrowing down to the specific diagnostic methods and clinical implications. At the end of the introduction is stated the purpose of this review paper.

Clinical Evaluation and Laboratory Evaluation of Patients with Thyroid Nodules: The first sentence was split in two as asked. At line 81 of multinodular goiter, we performed all the changes that the reviewer suggested. As the reviewer asked the discussion paragraph of the nodule size, was put and integrated, before the presentation of the challenges of the multinodular goiter. Ring down artifact was explained as the reviewer rightly asked. The reviewer noted about line 130 the start of malignant features with microcalcifications and calcifications – the reason we preferred that is due to the high specificity and PPV.

  1. US Classification Systems: We changed line 238 as asked and put “to produce a numeric scoring“. Line 252 as asked we changed to “regularly used.” Regarding the Korean Society of Thyroid Radiology and European Thyroid Association TIRADS we did not mention studies assessing the above classifications as we have done with the other more important and more used classification systems, because that would add weight and slow down the fluency of reading the paper for the general US practitioner.
  2. FNAB: 395 - the sentence was split it into two as asked by the reviewer. As the reviewer suggested, the Italian consensus was briefly explained and compared it with the Bethesda system. However it is intended by the authors as marginal in the context of the paper and of the paragraph. Regarding Hurthe cells we gave explanations as rightly asked and restructured the paragraph. Changes were made at the two sentences with lymphoma and anaplastic carcinoma.

Round 2

Reviewer 1 Report

Comments and Suggestions for Authors

David et al. improve the paper with corrections suggested by the reviewer.

Accept in present form.

Thank you for calling me to review the paper for Biomedicines.

Author Response

I learn that no other comments were made, and I thank you for all the suggestions received in Round 1 which made our paper better.

Reviewer 3 Report

Comments and Suggestions for Authors

The authors have attended all comments of this reviewer.

Author Response

(The authors gave the same response as above.)

Reviewer 4 Report

Comments and Suggestions for Authors

The authors significantly improved the manuscript.  No futher comments.

Comments on the Quality of English Language

Very minor corrections are needed

Author Response

(The authors gave the same response as above.)
